

# An adaptive congestion control algorithm for improving Transmission Control Protocol performance over cellular-to-cloud networks

Omar Imhemed Alramli[1,2], Zurina Mohd Hanapi[2], Mohamed Othman[2,3], Idawaty Ahmad[2] and Normalia Samian[2]

[1] Department of Telecommunications and Networking, Misurata University, Misurata, Libya
[2] Department of Communication Technology and Network, Universiti Putra Malaysia, Serdang, Selangor, Malaysia
[3] Laboratory of Computational Science and Mathematical Physics, Institute for Mathematical Research (INSPEM), Universiti Putra Malaysia, Serdang, Selangor, Malaysia

## ABSTRACT

In recent years, significant advancements have been made in enhancing congestion control algorithms (CCAs) within the Transmission Control Protocol (TCP) for end-to-end communication in millimeter-wave (mmWave) cellular networks. However, TCP often struggles to effectively utilize available bandwidth in mmWave-based cellular-to-cloud networks due to round-trip time (RTT) constraints, resulting in suboptimal throughput and latency that negatively impact cellular-to-cloud applications. To address this limitation, we propose an adaptive CCA, MRVHS-based CCA, which integrates maximum segment size (MSS) and RTT variations into a comprehensive CCA framework. MRVHS is implemented and evaluated using the ns-3 network simulator, with its performance compared against well-established TCP variants, including NewReno, HighSpeed, CUBIC, bottleneck bandwidth and RTT (BBR), and fuzzy logic-based TCP (FB-TCP) in cellular-to-cloud networks. Simulation results indicate that MRVHS consistently achieves higher average throughput while maintaining low latency across varying packet error rate (PER) levels. Notably, MRVHS significantly outperforms HighSpeed and CUBIC at the highest PER, achieving a 4.75% improvement over BBR in high-PER scenarios. Moreover, MRVHS demonstrates substantial throughput gains in medium- and low-PER conditions, consistently surpassing the benchmark TCP protocols. Furthermore, MRVHS demonstrates substantial throughput gains under medium and low PER conditions, surpassing the benchmark TCP protocols.

# INTRODUCTION

Since the establishment of the Internet, congestion control (CC) has been a major challenge and has attracted significant research interest. Congestion control involves adjusting how much data a host sends to avoid overwhelming the network, utilizing

Corresponding authors
Omar Imhemed Alramli,
o_alrmli@yahoo.com
Zurina Mohd Hanapi,
zurinamh@upm.edu.my

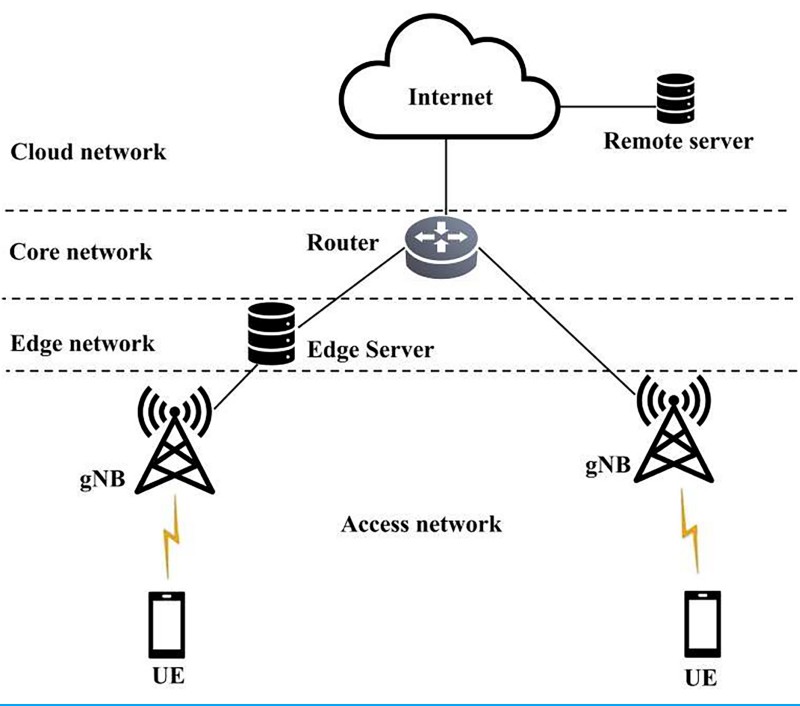

**Figure 1  Edge and remote server networks.**     

available bandwidth, and reducing latency to meet the requirements of mobile applications. The effectiveness of congestion control algorithms (CCA) can significantly impact network performance over cellular-to-cloud networks and the overall user experience (*Afanasyev et al., 2010*; *Niu et al., 2015*; *Zhang et al., 2017a*; *Mezzavilla et al., 2018*; *Haile et al., 2021*; *Lorincz, Klarin & Ožegović, 2021*; *Jeddou et al., 2023*; *Sudhamani et al., 2023*; *Biswal & Patel, 2024*).

Many congestion control mechanisms have been developed to improve end-to-end communication performance (*Wang et al., 2016*). However, the CCAs can behave unpredictably in wireless environments such as millimeter-wave (mmWave) cellular networks due to factors like mobility, handovers, interference, packet loss due to errors, and delays, especially in the presence of remote servers through the cloud (*Afanasyev et al., 2010*; *Alrshah et al., 2014*; *Mezzavilla et al., 2018*; *Poorzare & Augé, 2021b*; *Kim & Cho, 2022*; *Korbi, Zhani & Kaippallimalil, 2024*).

Naturally, cellular networks connect to servers, where the servers are located at the edge of a base station (*gNB*) or in a remote location through the Internet cloud, as shown in Fig. 1. This capability is crucial for seamless user experiences, as cellular network users often need to access applications hosted on servers in distant regions. Thus, ensuring that Transmission Control Protocol (TCP) can adapt to the remote server is essential for maintaining optimal high data rate, minimizing latency, and providing reliable connectivity (*Perdana et al., 2021*). This adaptability is vital in modern mobile applications that demand high data rates and low latency in mmWave cellular networks (*Zhang, Patras & Haddadi, 2019*; *Salahdine, Han & Zhang, 2023*; *Alramli et al., 2024*; *Islam et al., 2024*).

Therefore, a CCA is essential for managing data rates and latency to maintain the strength and effectiveness of cellular-to-cloud networks in meeting the demands of modern mobile applications.

In this article, we address the challenges of TCP performance in high-speed, lossy networks, particularly in mmWave cellular-to-cloud networks. Although CCAs have already been implemented in mmWave cellular networks, further research is still needed to enhance TCP performance, specifically in cellular-to-cloud environments, to meet the demands of the era's mobile applications. The motivations of this article are how MRVHS directly targets CCA's issues to attain high rates and low latency over cellular-to-cloud networks by integrating RTT variation (RTTV) and maximum segment size (MSS) metrics into the congestion window adaptation of HighSpeed TCP. Moreover, enhancing the responsiveness of the CCA in high-latency and lossy cellular-to-cloud networks. Furthermore, outperforming existing state-of-the-art CCAs such as HighSpeed, CUBIC, and BBR, especially in high-PER conditions and with remote servers. Given these considerations, this article presents the following key contributions.

- Introducing a new CCA, MRVHS-based CCA, specifically designed to address challenges in cellular-to-cloud networks and enhance TCP performance in these environments.
- To evaluate the proposed CCA performance in cellular-to-cloud networks. The performance of the proposed protocol is compared against five benchmark TCP protocols: NewReno (*Floyd, Henderson & Gurtov, 2004*), HighSpeed (*Floyd, 2003*), CUBIC (*Ha, Rhee & Xu, 2008*), bottleneck bandwidth and round-trip propagation time (BBR) (*Cardwell et al., 2016*), and fuzzy logic-based TCP (FB-TCP) (*Poorzare & Augé, 2021a*), under a range of mmWave cellular-to-cloud network conditions and different packet error rates (PERs) are considered in the simulation scenario.

The remainder of this article is organized as follows: 'Related Works' reviews the related work. Following that, 'MRVHS-Based CCA: the proposed algorithm' introduces and elaborates the proposed algorithm. Furthermore, 'Performance Evaluation of MRVHS-based CCA' provides the results and discussion, including a performance comparison of *cwnd* fluctuations, throughput, and latency to the benchmark TCP variants. Lastly, 'Conclusion' concludes the article with a summary of the findings and suggestions for future research.

## RELATED WORKS

To tackle the challenge of mmWave cellular networks on TCP performance, several TCP variants have been proposed as shown in Table 1, including D-TCP (*Kanagarathinam et al., 2018*), TCP-Drinc (*Xiao, Mao & Tugnait, 2019*), NexGen-TCP (*Kanagarathinam et al., 2020*), FB-TCP (*Poorzare & Augé, 2021a*), TCP-FLASH (*Guo & Lee, 2021*), mmS-TCP (*Kim & Cho, 2022*), RBBR (*Haile et al., 2022*), and Yinker-BBR (*Xie et al., 2023*). Despite the researchers' efforts, none of these TCP variants fully leverage the available bandwidth in cellular-to-cloud networks. In terms of achieving higher data rates and lower latency, the performance of these TCP variants remains suboptimal in mmWave

**Table 1  Overview of TCP protocols for mmWave cellular networks.**

| Ref. No. | Article title | Features | Compared schemes | Evaluation method | Limitations |
|---|---|---|---|---|---|
| *Polese, Jana & Zorzi (2017)* | Advanced 5G-TCP: Transport Protocol for 5G Mobile Networks | Presents 5G-TCP based on HSTCP protocol | Compared with HS TCP and Reno protocols | Simulation | Relies on predefined values, may not adapt to dynamic network conditions |
| *Xiao, Mao & Tugnait (2019)* | TCP-Drinc: Smart Congestion Control Based on Deep Reinforcement Learning | Uses deep reinforcement learning for congestion control | Evaluated against NewReno, CUBIC, Hybla, Vegas, and Illinois | Simulation | Depends on historical data, which may be less effective in rapidly changing conditions |
| *Kanagarathinam et al. (2020)* | NexGen D-TCP: Next Generation Dynamic TCP Congestion Control | Dynamically estimates bandwidth to adjust cwnd | Compared with BBR, CUBIC, Reno, Westwood, Tahoe, CLTCP | Simulation and live experiment | Overestimates bandwidth, which can lead to potential inaccuracies in cwnd adjustment. |
| *Poorzare & Augé (2021a)* | FB-TCP: A 5G mmWave Friendly TCP for Urban Deployments | Fuzzy-based Algorithm for urban development challenges | Compared with NewReno, HighSpeed, CUBIC, BBR | Simulation | Degrades in performance with small buffer sizes |
| *Haile et al. (2022)* | RBBR: A Receiver-Driven BBR in QUIC for Low-Latency in Cellular Networks | Uses Kalman filter for bandwidth estimation and employs receiver-driven rate estimation | Compared with CUBIC and BBR | Testbed | High complexity and increased overhead |
| *Xie et al. (2023)* | Yinker: Adaptive BBR for High-Throughput, Low-Latency Data Transmission | Adapts pacing gain in BBR based on network conditions | Evaluated against CUBIC and BBR over Wi-Fi and 5G networks | Simulation and testbed | Lacks proof for queuing delay threshold expansion based on ($RTT$). |
| *Netalkar, Chen & Raychaudhuri (2023)* | mmCPTP: Cross-Layer Pull-Based Transport for 5G mmWave | Optimizes for 5G mmWave with cross-layer, pull-based data transmission | Compared with NewReno and CUBIC | Simulation | Performance reliant on cross-layer information, complex implementation |

cellular networks, primarily due to the constraints of cellular-to-cloud networks. These limitations include difficulties with obstacle penetration, which can lead to non-line-of-sight (*NLoS*) conditions between the *gNB* and the user equipment (*UE*). Additionally, the delay variations caused by the server's location at a remote deployment affect TCP performance (*Mezzavilla et al., 2018; Zhang et al., 2019; Jeddou et al., 2023*). Thus, legacy TCPs encounter issues with round-trip time (*RTT*) fluctuations over cellular-to-cloud networks. In such cases, performance can drop significantly due to *RTT* variations, which hinder the TCP protocol's CCA in accurately adjusting the cwnd size, impacting key network performance metrics such as throughput and latency.

To summarize, the legacy CCAs issues in cellular-to-cloud networks are.

- The inefficiency of TCP's CCAs over mmWave cellular-to-cloud networks, especially under fluctuating RTT conditions due to the presence of remote servers.
- The slow congestion window size (cwnd) growth in traditional loss-based when RTT is high or variable.
- The inability of existing CCAs to fully utilize bandwidth and maintain low latency under high PER and NLoS conditions over cellular to cloud networks.

## MRVHS-BASED CCA: THE PROPOSED ALGORITHM

In the congestion avoidance (CA) phase of loss-based algorithms, CCA requires approximately one $RTT$ to increase the $cwnd$ by one segment. On the other hand, $RTT$-dependent CCAs adjust their $cwnd$ by one for each $RTT$ during the CA phase. Consequently, if the $RTT$ is short, the increase of $cwnd$ occurs faster; but for longer $RTT$, the $cwnd$ increase becomes much slower. However, in cellular-to-cloud networks, where $RTT$s are often longer and fluctuating, results in a significantly slow increase in $cwnd$, which affects TCP performance in terms of throughput and latency. To overcome $RTT$ variation issue in the presence of a remote server on the cloud, $RTT$ variation alongside $MSS$ is incorporated into the $cwnd$ calculation of HighSpeed CCA. Specifically, Eq. (1) is used where $RTT_{max}$ represents the maximum $RTT$, $RTT_{base}$ is the minimum $RTT$, and $RTT_{cur}$ refers to the recent $RTT$ derived from the latest acknowledgment (ACK). The other parameter considered alongside $RTT$ variations to calculate the $cwnd$ size is the magnitude of the maximum segment size ($MSS$) as shown in Eqs. (2) and (3), respectively. Consequently, instead of updating the $cwnd$ by preset value, the MRVHS-based CCA updates its $cwnd$ during the congestion avoidance (CA) stage, as outlined in Eqs. (4) and (5), respectively. On the other hand, MRVHS decreases the $cwnd$ whenever three duplicate acknowledgments (3DACKs) are received or retransmission-time out ($RTO$) is triggered, following the HighSpeed CCA mechanism as shown in Eq. (6). For instance, in HighSpeed CCA, the cwnd is adjusted by two factors, where $\alpha(cwnd)$ increases the cwnd and $\beta(cwnd)$ decreases the cwnd. when the $cwnd$ is less than or equal to 38 packets, $\alpha(cwnd) = 1$ and $\beta(cwnd) = 0.5$. As $cwnd$ increases beyond 84 packets, $\alpha(w)$ can grow up to 70, while $\beta(w)$ decreases to 0.1 (Afanasyev et al., 2010).

$$Time\ Ratio = (RTT_{max} + RTT_{base})/RTT_{cur} \tag{1}$$
$$One\ segment\ size = segment\ size/number\ of\ segments \tag{2}$$
$$MSS = segment\ size/number\ of\ segments \tag{3}$$
$$\alpha = \sqrt{MSS \times Time\ Ratio} \tag{4}$$
$$cwnd = cwnd + (\alpha/cwnd) \tag{5}$$
$$cwnd = cwnd - (\beta/cwnd). \tag{6}$$

To improve the TCP performance over mmWave cellular-to-cloud networks involving a remote server, the proposed MRVHS-based CCA integrates both $RTT$ variations and $MSS$ mechanisms into HighSpeed CCA. Thus, Fig. 2 presents a comparison of the $cwnd$ between MRVHS and HighSpeed and CUBIC CCAs. Additionally, Fig. 2 shows that MRVHS effectively utilizes the available bandwidth through its novel mechanism, optimizing the $cwnd$ to enhance TCP performance to improve the throughput over cellular-to-cloud networks. Furthermore, Algorithm 1 outlines the functionality of the MRVHS CCA based on Eqs. (1)–(6) respectively. For instance, $RTT$ variation allows the algorithm to dynamically adapt to changing network delays, especially in cellular-to-cloud environments, where $RTT$ is often high and fluctuating due to the presence of remote servers and mobility (e.g., LoS/NLoS transitions). Moreover, MSS-awareness improves

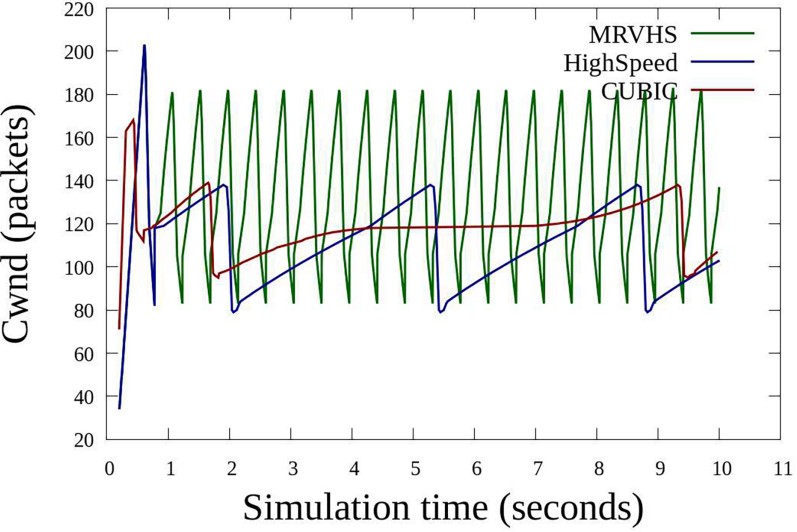

**Figure 2** **MRVHS cwnd increase *vs.* HighSpeed and CUBIC.**

granularity in cwnd updates, ensuring that the cwnd growth is better aligned with the MSS, which leads to more accurate and scalable congestion control. Furthermore, by integrating both *RTT* and *MSS*, MRVHS responds more efficiently to network dynamics than traditional CCAs like HighSpeed or CUBIC, which rely on fixed growth functions or less adaptive pacing mechanisms.

# PERFORMANCE EVALUATION OF MRVHS-BASED CCA

To evaluate the performance of the proposed CCA, the proposed MRVHS module is integrated into the ns-3 network simulator, and extensive simulation experiments are conducted. The effectiveness and efficiency of MRVHS are evaluated over cellular-to-cloud networks involving a remote server scenario. The following subsections provide detailed information about the simulation setup, including the parameters used, the network topology, and the specific conditions under which the experiments were performed.

## Experiment setup

Extensive simulation experiments are conducted using the well-known ns-3 network simulator to evaluate the proposed CCA by comparing its performance to several state-of-the-art CCAs, including NewReno, HighSpeed, CUBIC, BBR, and FB-TCP. The 5G mmWave networks are developed based on the LENA framework, an extension of ns-3, further expanded in the ns-3 mmWave module. The implementation of the channel model is presented in *Zhang et al. (2017b)*, while support for dual connectivity is discussed in *Polese et al. (2017)*, *Polese, Mezzavilla & Zorzi (2016)*. A detailed overview of the ns-3 mmWave module can be found in *Mezzavilla et al. (2018)*. This module serves as a robust and accessible simulation platform for modeling various elements of 5G mmWave

**Algorithm 1    MRVHS-based CCA.**

1   Initialize: *cwnd, ssthresh, β, RTTbase, RTTcur, RTTmax*;

2   **if** *there is data to send* **then**

3        **if** *RTO not expired* **then**

4              **if** *no 3DACK* **then**

5                **if** *cwnd < ssthresh* **then**

6                      $cwnd = cwnd + 1$;

7                **else**

8                      $RTT_{cur} = Ack_{time} - Send_{time}$;

9                      **if** $RTT_{base} > RTT_{cur}$ **then**

10                          $RTT_{base} = RTT_{cur}$;

11                     **end**

12                     **if** $RTT_{max} < RTT_{cur}$ **then**

13                          $RTT_{max} = RTT_{cur}$;

14                     **end**

15                     $TimeRatio = (RTT_{max} + RTT_{base})/RTT_{cur}$;

16                     $MSS = segment\ size/number\ of\ segments$;

17                     $\alpha = \sqrt{MSS \times Time\ Ratio}$;

18                     $cwnd = cwnd + \alpha/cwnd$;

19              **end**

20        **else**

21              $cwnd = cwnd - \beta/cwnd$;

22        **end**

23    **else**

24        Slow start stage;

25    **end**

26 **end**

networks, including protocol layers and channel behaviors in line with 3GPP specifications. A key feature of this module is its integration with direct code execution (DCE) (*Tazaki et al., 2013*), enabling the use of the native Linux TCP/IP stack as the networking stack for ns-3 nodes. Additionally, it supports simulation across a wide range of frequencies—from 6 GHz up to 100 GHz—based on the official 3GPP channel models (*Poddar et al., 2023*).

As illustrated in Fig. 3, the network consists of UE communicating with a data center server through the cloud. The UE connects to the network *via* cellular networks, using a gNB station. To simulate a cellular-to-cloud environment involving a remote server, the one-way propagation delay from the gNB to the server was set to 40 ms, resulting in a minimum RTT of 80 ms.

In this simulation scenario, the UE moves in and out of line of sight (LoS) conditions with the gNB station, transitioning to NLoS due to simulated obstacles such as buildings.

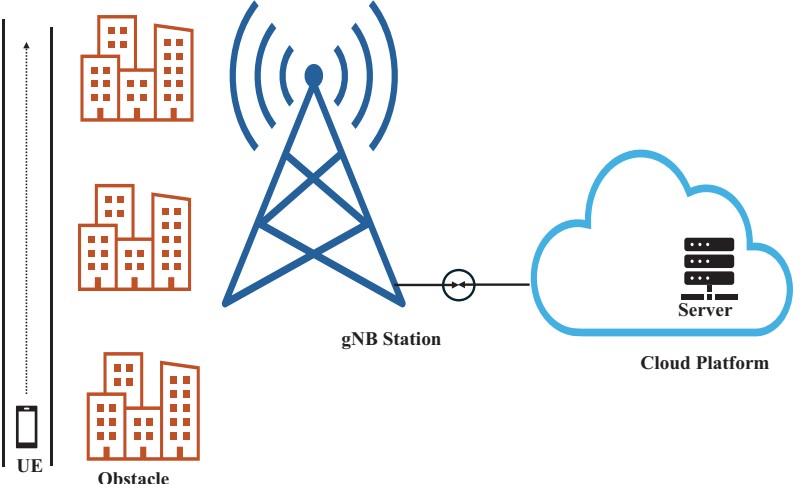

**Figure 3  Network topology.**

The UE stops briefly at certain points, then continues moving through both LoS and NLoS conditions, stops again for a short duration, and finally moves away from the gNB station before stopping. As a result, this scenario is designed to investigate the effects of transitioning between LoS and NLoS, the impact of stopping and resuming movement, and the influence of distance between the UE and the gNB station on CCA performance over cellular-to-cloud networks.

The simulation parameters used in these experiments are crucial for accurately reflecting real-world conditions. The carrier frequency is set to 28 GHz to simulate high-frequency 5G communication. Moreover, a building obstacle propagation loss model is incorporated to account for signal attenuation caused by environmental obstacles. Furthermore, the communication bandwidth is configured at 1 GHz, enabling high data transmission rates. In addition, the retransmission timeout (RTO) is fixed at 1 s, while various PER are introduced, ranging from no PER to increasingly lossy conditions to simulate different levels of network reliability. Each simulation runs for 60 s, providing sufficient time to observe the CCAs' behavior and ensure the behavior reaches a steady state under dynamic cellular network conditions.

Furthermore, these carefully chosen parameters help ensure that the experiments capture the realistic challenges of cellular-to-cloud communications and enable accurate comparisons between the proposed CCA and existing ones.

## RESULTS AND DISCUSSION

This section analyzes the behavior of the MRVHS-based CCA compared to benchmark CCAs. It also presents performance results on cwnd fluctuations, throughput, and latency, illustrating how PER under RTT variations affects overall performance in a remote server scenario over a cellular-to-cloud network.

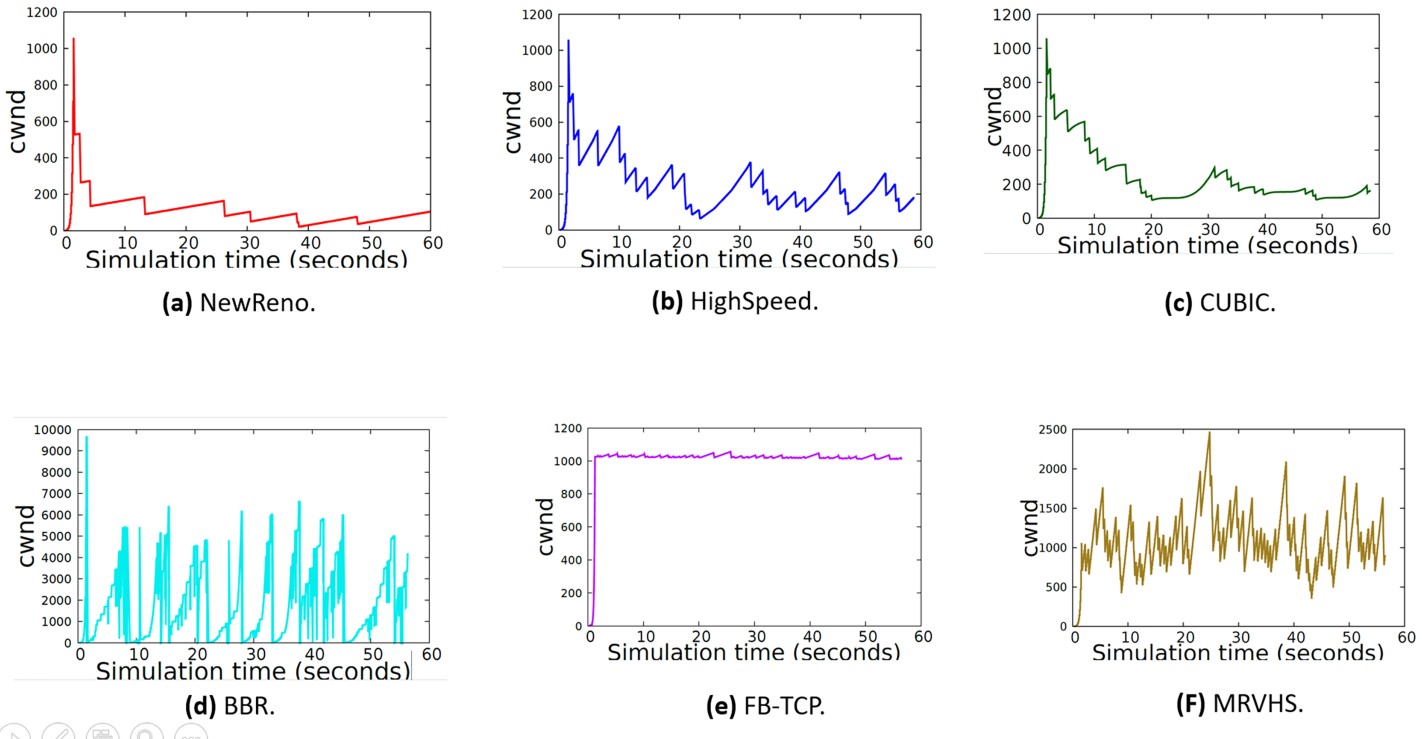

**Figure 4 Cwnd analysis of the protocols *vs*. high PER.** (A) NewReno; (B) HighSpeed; (C) CUBIC; (D) BBR; (E) FB-TCP; (F) MRVHS.

## Congestion window fluctuations

The objective of adjusting the congestion window (cwnd) is to manage congestion more effectively, maximizing throughput while maintaining or reducing latency. Thus, the estimated in-flight data must be smaller than both the sender's congestion window size and the receiver's window size. Therefore, the updated sender window size should be adjusted to the minimum of these two values. A suitable congestion window size and fewer loss events indicate better congestion control, which, in turn, leads to a higher data rate, to avoid bufferbloat phenomena, and more efficient utilization of the network's available bandwidth.

To compare the proposed protocol with the state-of-art protocols, Figs. 4, 5, 6, and 7 show the cwnd fluctuations of CCA variants against different PERs. The figures illustrate the behavior of the cwnd for different CCAs over simulation time under remote server scenarios with varying conditions, such as mobility, stopping, walking, and driving. These conditions mimic LoS and NLoS situations, as well as changing distances between the UE and the gNB station to simulate signal attenuation conditions.

When a high PER is considered, MRVHS-TCP demonstrates the highest cwnd magnitude, varying from 500 to 2,000 packets in high PER scenarios, as shown in Fig. 4, whereas HighSpeed's cwnd ranges from 100 to 400 packets, as depicted in Fig. 4B. In

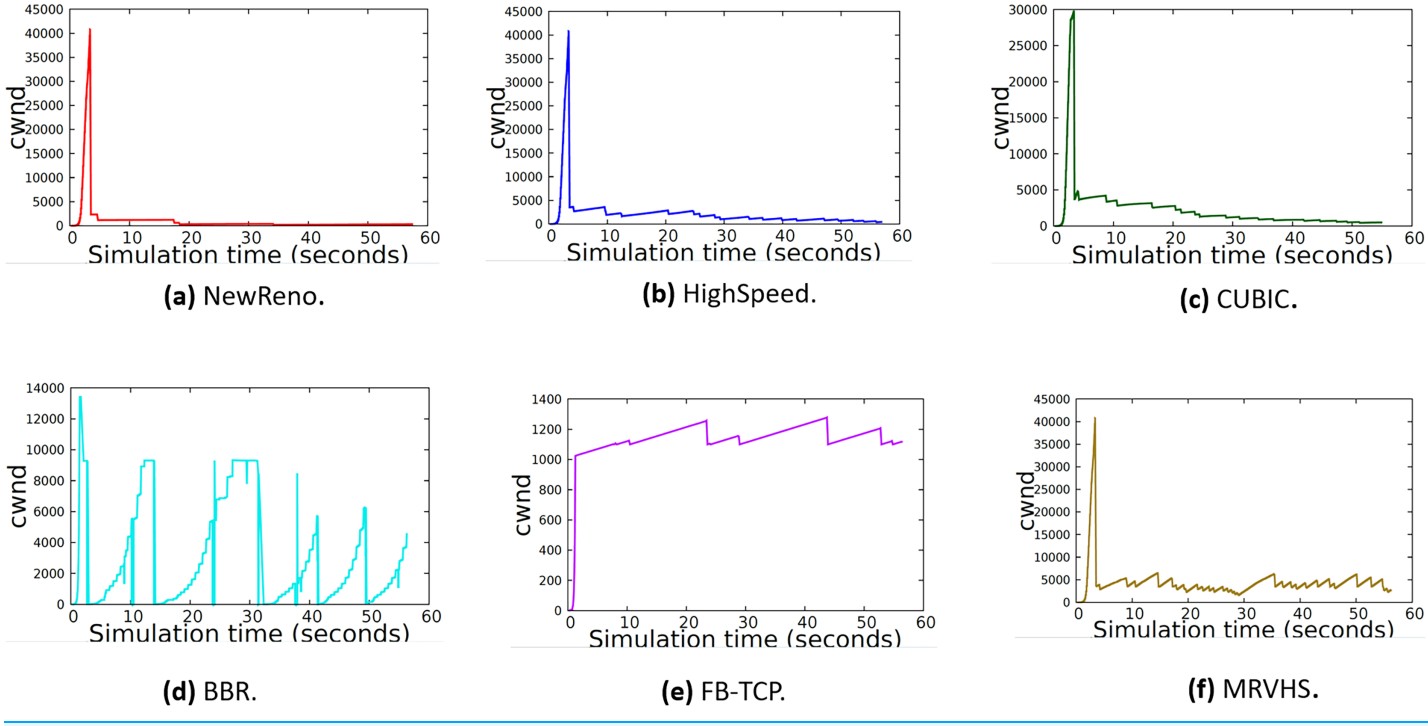

**Figure 5 Cwnd analysis of the protocols *vs*. medium PER.** (A) NewReno; (B) HighSpeed; (C) CUBIC; (D) BBR; (E) FB-TCP; (F) MRVHS.

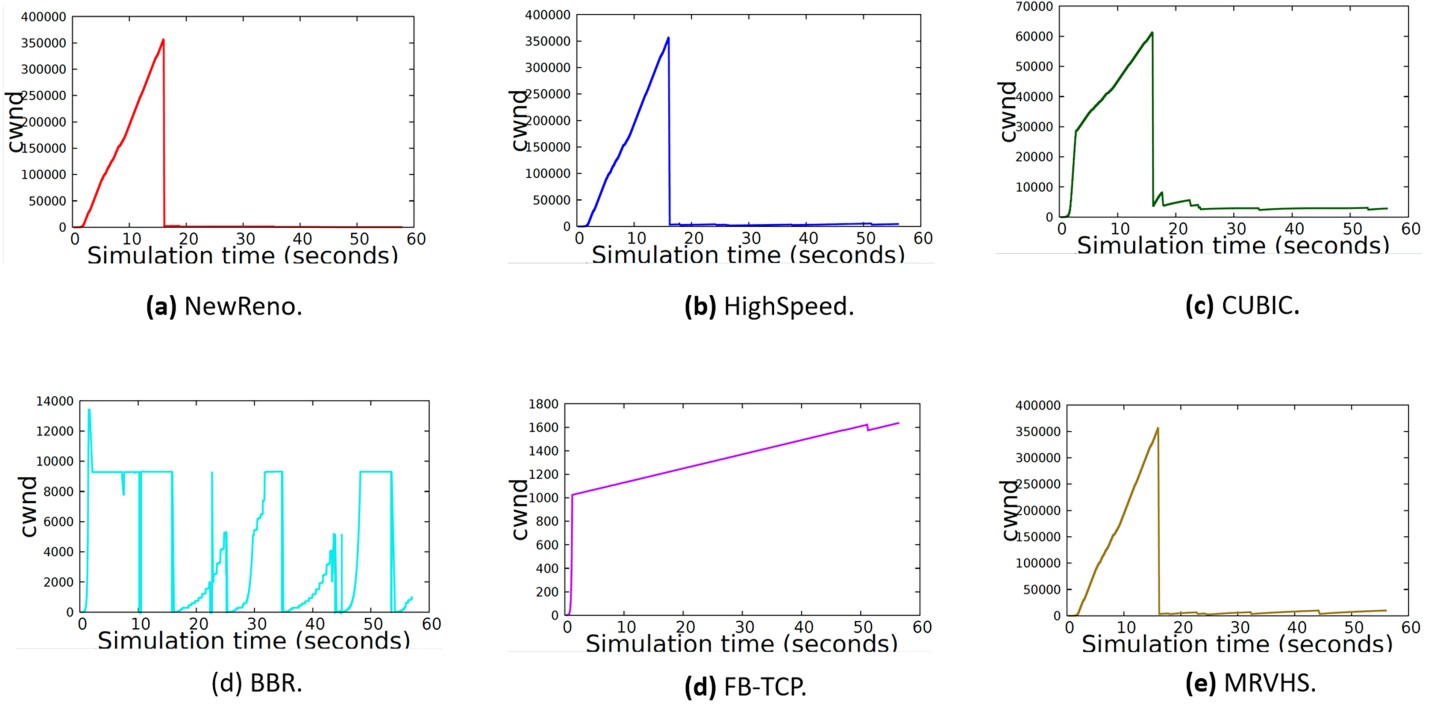

**Figure 6 Cwnd analysis of the protocols *vs*. small PER.** (A) NewReno; (B) HighSpeed; (C) CUBIC; (D) BBR; (E) FB-TCP; (F) MRVHS.

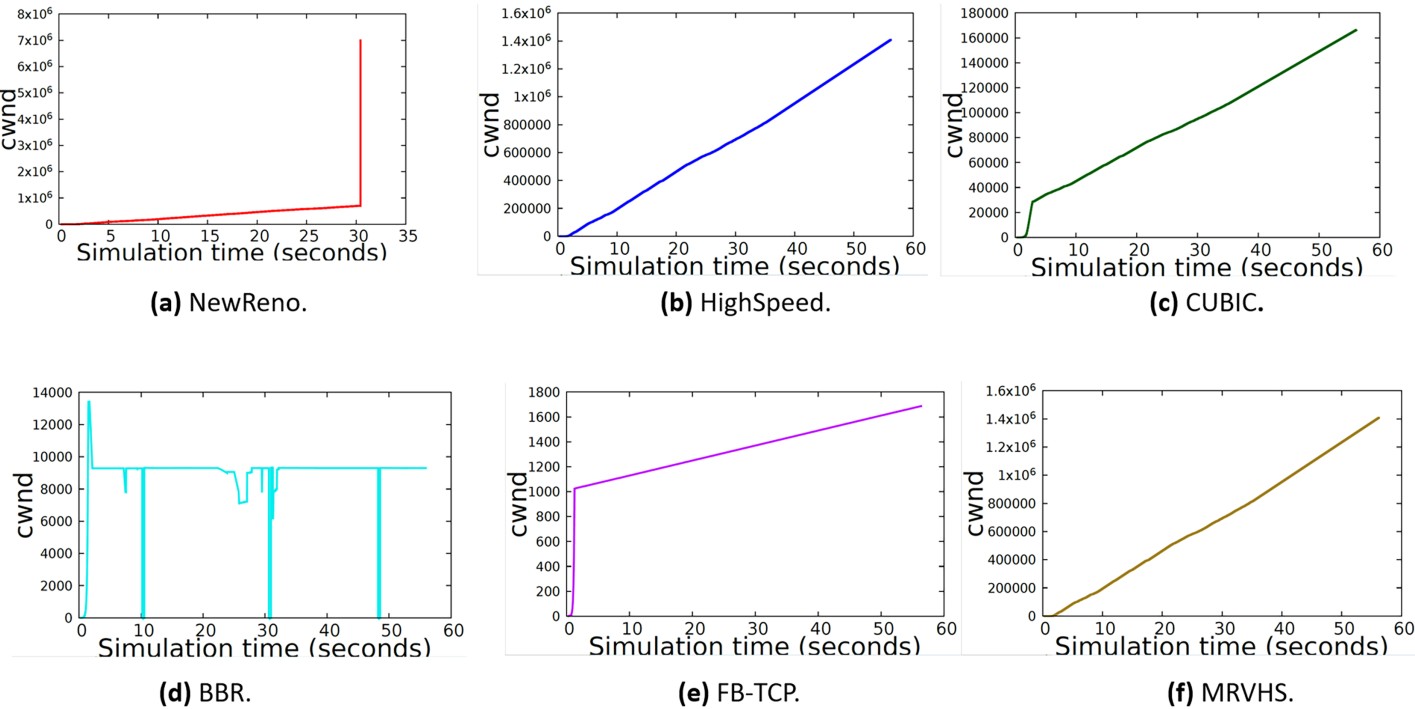

**Figure 7 Cwnd analysis of the protocols vs. null PER.** (A) NewReno; (B) HighSpeed; (C) CUBIC; (D) BBR; (E) FB-TCP; (F) MRVHS.

contrast, Fig. 4A shows that the cwnd of NewReno is limited to between 100 and 200 packets, while Fig 4C. indicates that the cwnd of CUBIC hovers around 300 packets. The smaller cwnd sizes in loss-based CCAs are attributed to their sensitivity to packet loss, particularly in the presence of high PER in the network. However, the cwnd of BBR fluctuates between zero and 5,000 packets, as shown in Fig. 4D while FB-TCP cwnd swings around 1,000 packets as shown in Fig. 4E.

The cwnd results indicate that MRVHS CCA adjusts the cwnd to values distinct from the lower cwnd typical of loss-based CCAs and the higher cwnd seen in BBR. MRVHS outperforms other TCP variants in terms of bandwidth utilization in the worst-case high PER scenario over cellular-to-cloud networks with remote servers. This performance is attributed to the combination of RTTV and MSS mechanisms, which provide MRVHS with the flexibility to handle data transmission effectively in remote server scenarios over cellular-to-cloud networks, resulting in higher throughput and better utilization of the available bandwidth.

## The throughput

Throughput refers to the data rate achieved between end-to-end hosts. The throughput results for various PER scenarios demonstrate that MRVHS-TCP consistently achieves high average throughput across all cases when a remote server is employed in the cellular-

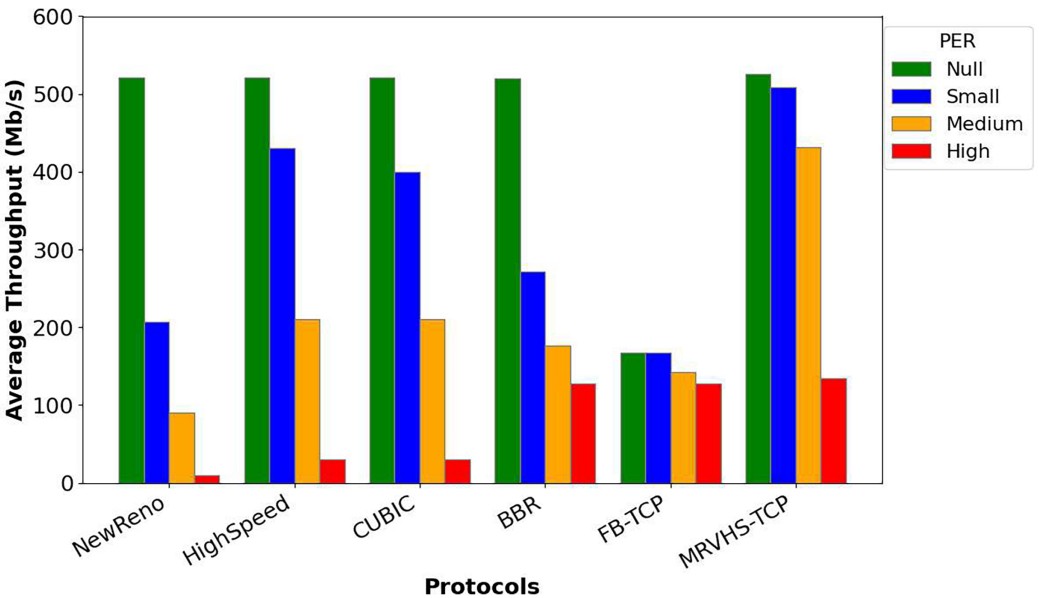

**Figure 8 Average throughput of TCP variants *vs*. different PERs.**

**Table 2 MRVHS throughput improvement (%) compared to the benchmark protocols.**

| PER | TCP protocols | | | | |
|---|---|---|---|---|---|
| | **NewReno** | **HighSpeed** | **CUBIC** | **BBR** | **FB-TCP** |
| Null | 0.89 | 0.89 | 0.89 | 1.14 | 213.70 |
| Small | 146.01 | 18.42 | 27.31 | 87.19 | 205.06 |
| Medium | 379.02 | 105.29 | 105.29 | 144.64 | 202.53 |
| High | 1,242.05 | 347.35 | 347.35 | 4.75 | 5.48 |

to-cloud network. As shown in Fig. 8 and Table 2, MRVHS outperforms all five competing CCAs across all PER scenarios. It achieves the highest average throughput, followed by HighSpeed, whereas the performance of other CCAs is significantly lower, particularly that of FB-TCP in the zero and small PER scenarios. Additionally, Fig. 9 illustrates the instantaneous throughput of TCP variants over the simulation time of the experiment. As PER increases, the advantage of MRVHS becomes increasingly apparent due to the functionality of the MRVHS-based mechanism. Notably, throughput increases directly with changes in the congestion window size. Thus, integrating the attributes of RTTV and MSS techniques into the HighSpeed CCA is highly beneficial and promising for implementation in real operating systems. Overall, MRVHS effectively meets the requirements for high data rates and low latency in mobile applications over cellular-to-cloud networks.

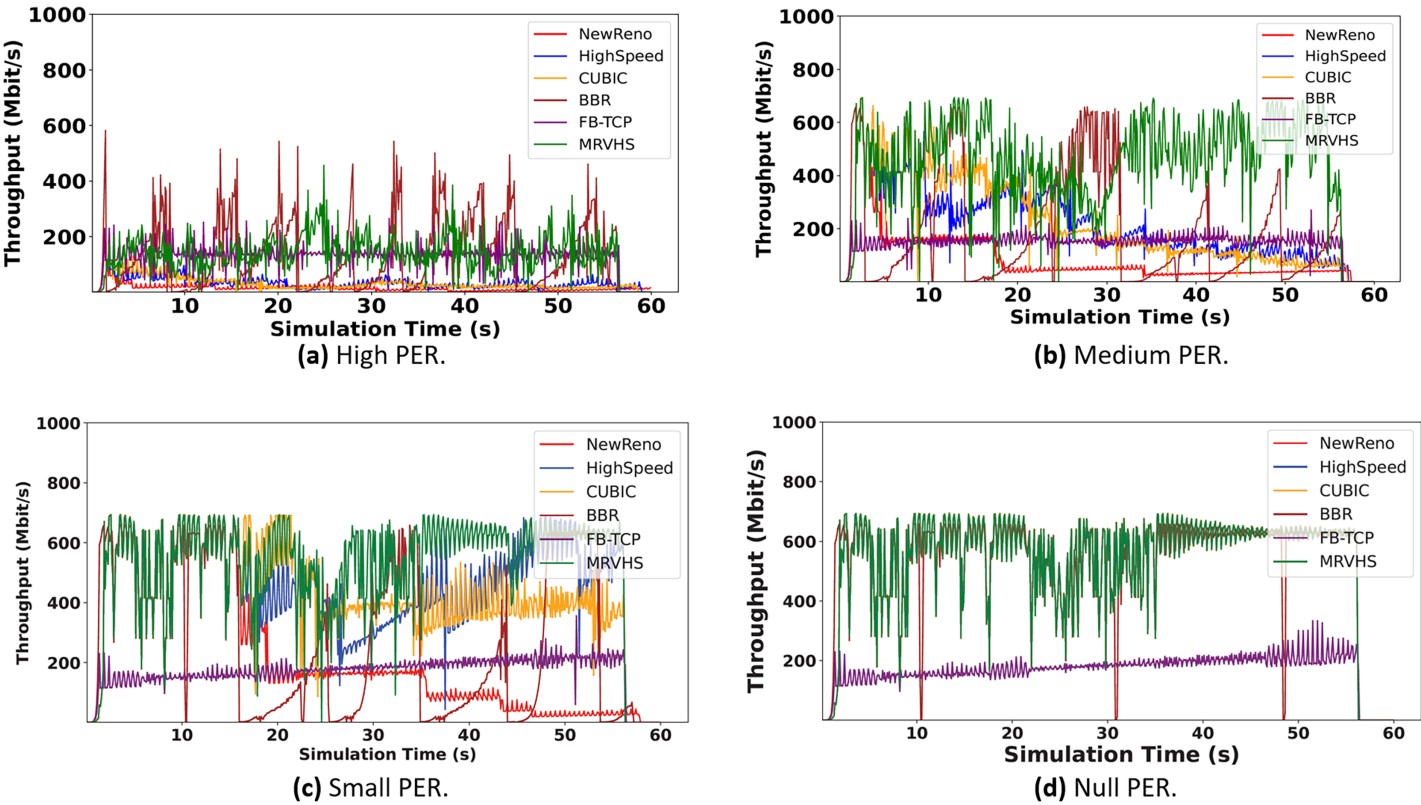

**Figure 9 Instantaneous throughput comparison.** (A) High PER; (B) Medium PER; (C) Small PER; (D) Null PER.

## The latency

Latency refers to the time required for a data packet to travel from the source to the destination. Several factors influence latency, including the nature of the transmission medium (*e.g.*, wired, wireless, or satellite) and the geographical distance between the sender and receiver. To calculate the average latency, all delays encountered during transmission are summed and then divided by the number of measurements. Among these factors, propagation delay is a key contributor to overall latency.

The variations in delay between the UE and the remote server over a cellular-to-cloud network, combined with the mobility of the UE from LoS to NLoS, the cwnd decreases. This reduction in the cwnd leads to a decline in the data rate. As a result, data packets begin to accumulate in the buffer, while TCP remains unaware of the primary cause of packet loss. Consequently, this increased buffer occupancy increases latency and degrades overall network performance.

Despite these challenges, MRVHS shows significant advantages. It achieves higher throughput compared to other TCP variants across various scenarios with different PERs while maintaining latency levels comparable to most other protocols. As illustrated in Fig. 10, MRVHS exhibits only slight differences in latency compared to loss-based and

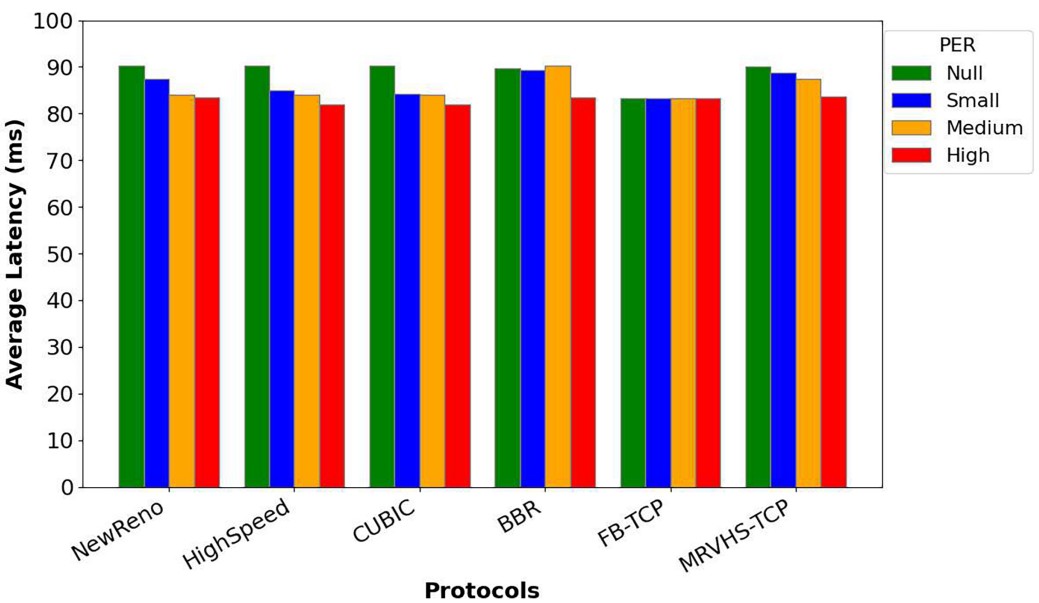

**Figure 10 Average latency of TCP variants *vs*. different PERs.**

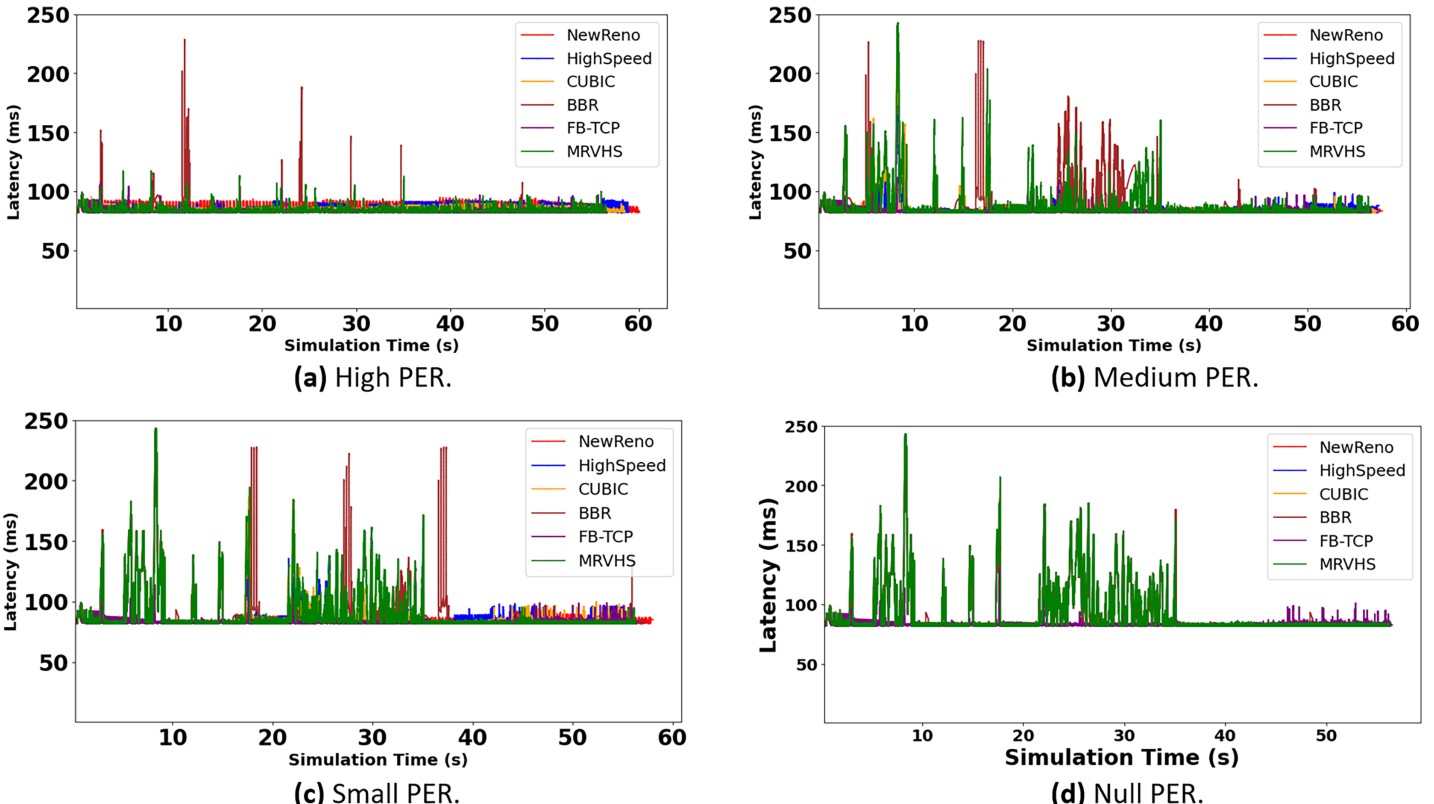

**Figure 11 Instantaneous latency comparison.** (A) High PER; (B) Medium PER; (C) Small PER; (D) Null PER.

**Table 3  MRVHS latency improvement (%) compared to the benchmark protocols.**

| PER | TCP protocols | | | | |
|---|---|---|---|---|---|
| | **NewReno** | **HighSpeed** | **CUBIC** | **BBR** | **FB-TCP** |
| Null | −0.27 | −0.27 | −0.27 | 0.38 | 8.06 |
| Small | 1.36 | 4.34 | 5.21 | −0.72 | 6.41 |
| Medium | 4.14 | 4.14 | 4.14 | −3.11 | 4.98 |
| High | 0.20 | 2.04 | 2.04 | 0.30 | 0.50 |

fuzzy-based protocols, as seen in Fig. 11 and Table 3. Notably, MRVHS even achieves lower latency in some instances, particularly in medium and small PER scenarios, when compared to BBR. These results highlight the protocol's effectiveness and suitability for applications requiring high speed and minimal delay, especially when interacting with remote servers over cellular-to-cloud networks.

# CONCLUSION

An adaptive CCA named MRVHS is proposed in this work and designed for cellular-to-cloud networks. We evaluated its performance through ns-3 simulation-based experiments, benchmarking it against established TCP protocols, including NewReno, HighSpeed, CUBIC, BBR, and FB-TCP. The results demonstrated that MRVHS achieved superior bandwidth utilization and reduced latency compared to these existing protocols. Overall, MRVHS outperforms the compared TCP protocols in terms of both average throughput and latency in remote server scenarios within cellular-to-cloud networks. For instance, In the presence of higher *PER*s, MRVHS achieves 1,242.05%, 347.35%, 347.35%, 4.75%, and 5.48% greater average throughput performance compared to NewReno, HighSpeed, CUBIC, BBR, and FB-TCP, respectively. These findings suggest that MRVHS could be a promising candidate for use alongside benchmark protocols such as CUBIC and BBR, supporting implementation across different operating systems to accommodate a wide range of network environments. For future work, we aim to investigate the performance of the proposed protocols across diverse network topologies and varying traffic conditions, and evaluate the fairness of MRVHS in comparison with state-of-the-art protocols.

## Funding
The authors received no funding for this work.

## Competing Interests
The authors declare that they have no competing interests.

## Author Contributions

- Omar Imhemed Alramli conceived and designed the experiments, performed the experiments, analyzed the data, performed the computation work, prepared figures and/or tables, authored or reviewed drafts of the article, and approved the final draft.
- Zurina Mohd Hanapi conceived and designed the experiments, performed the experiments, authored or reviewed drafts of the article, supervision, and approved the final draft.
- Mohamed Othman conceived and designed the experiments, performed the experiments, authored or reviewed drafts of the article, supervision, and approved the final draft.
- Idawaty Ahmad analyzed the data, authored or reviewed drafts of the article, supervision, and approved the final draft.
- Normalia Samian analyzed the data, authored or reviewed drafts of the article, supervision, and approved the final draft.

## Data Availability

The source code is available in the Supplemental File.

## Supplemental Information

Supplemental information for this article can be found online at http://dx.doi.org/10.7717/peerj-cs.2956#supplemental-information.

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
