# Peer review of "An adaptive congestion control algorithm for improving Transmission Control Protocol performance over cellular-to-cloud networks"

_PeerJ Computer Science, doi:10.7717/peerj-cs.2956_

## Round 0.1 · original submission · Major Revisions

You should carefully address the comments of the two major revision requests, particularly those regarding the algorithm's adaptability and more details of the proposed approach.

·

Basic reporting

This paper proposed an adaptive congestion control algorithm for improving TCP performance over cellular-to-cloud networks. However, these are some general comments.
-The abbreviation is introduced but not consistently used throughout the manuscript.
-Make sure that once an abbreviation is defined, it is used consistently.
-The issues addressed by the new algorithm are not sufficiently prominent.
-Please provide a detailed explanation of the motivation behind it.

Experimental design

An experimental study is sufficiently conducted. The following comments need to be addressed:
-The selection of tuning of parameters β is not adequately discussed.
-To provide more details on how these parameters are chosen and their impact on performance.
-The manuscript lacks a discussion of the algorithm’s limitations under varying network conditions.

Validity of the findings

The following results comparison should be incorporated:
- A fairness study of the proposed algorithm should be presented.
- Explain the advantages of integrating RTT variations and MSS into HighSpeed CCA, particularly in comparison to existing approaches.

Additional comments

The relevant publications may be cited, including the following papers:
i) A bandwidth delay product based modified Veno for high-speed networks: BDP-Veno (2024). Journal
of Network and Computer Applications, 231, p.103983.
ii) https://doi.org/10.1016/j.jnca.2016.03.018

Reviewer 2 ·

Basic reporting

The authors have presented the manuscript titled “An Adaptive Congestion Control Algorithm for Improving TCP Performance over Cellular-to-Cloud Networks.” While the overall presentation quality of the paper is commendable, the reviewer has the following concerns that need to be addressed:
• In the abstract, the authors state: “TCP often struggles to effectively utilize available bandwidth in mmWave-based cellular-to-cloud networks due to Round-Trip Time (RTT) constraints, resulting in suboptimal throughput and latency that negatively impact cellular-to-cloud applications”. However, the manuscript lacks a clear and detailed justification for how RTT constraints specifically affect TCP performance in these networks. The authors should provide theoretical explanations, empirical evidence, or references to support this claim.
• The introduction section is too brief and does not adequately justify the need for the proposed method. A clear motivation for the study and a detailed problem statement are missing. The authors should expand this section by: Explaining the significance of congestion control in cellular-to-cloud networks. Highlighting the gaps in existing approaches that the proposed method aims to address. Providing contextual examples or scenarios where the proposed algorithm would be beneficial.
• The Related Work section lacks coverage of recent research. The authors should incorporate more recent studies (preferably from the past five years) that focus on TCP performance in mmWave networks, congestion control algorithms, and cellular-to-cloud communication. Provide a comparative analysis of existing algorithms and clearly outline how the proposed method differs from or improves upon them.
• The Proposed Method section does not clearly explain the approach, its parameters, and the algorithmic steps involved. Equation 6 introduces a variable (β), which is not defined or explained anywhere in the manuscript. The authors must provide a detailed explanation of this parameter, including its role and impact on the algorithm. CWND Reduction Analysis: The proposed approach involves reducing the Congestion Window (CWND) using Equation 6, but there is no impact analysis provided. Discuss the implications of CWND reduction on TCP throughput, latency, and fairness. Provide analytical insights or simulation results that justify this approach.

Experimental design

The experimental setup section lacks critical details, making it difficult to replicate the study. The authors should clearly describe the simulation environment, tools used (e.g., NS-3), network parameters, and evaluation metrics. Refer to relevant studies to ensure that the experimental design aligns with established practices in the field.
• The results and discussion section does not provide a comprehensive analysis of the findings. Explain why the proposed method achieves better performance in the given scenarios. Provide comparisons with baseline methods and discuss the significance of performance gains in terms of throughput, latency, and packet loss. Discuss any limitations of the proposed approach and suggest future research directions.

Validity of the findings

• The results and discussion section does not provide a comprehensive analysis of the findings. Explain why the proposed method achieves better performance in the given scenarios. Provide comparisons with baseline methods and discuss the significance of performance gains in terms of throughput, latency, and packet loss. Discuss any limitations of the proposed approach and suggest future research directions.

Additional comments

Paper need significant improvement.

Reviewer 3 ·

Basic reporting

Report attached

Experimental design

Design or model needed

Validity of the findings

Author carried out simulation and result not clear

Annotated reviews are not available for download in order to protect the identity of reviewers who chose to remain anonymous.

Reviewer 4 ·

Basic reporting

This paper presents a novel adaptive congestion control algorithm (MRVHS-CCA) that significantly enhances TCP performance in mmWave cellular-to-cloud networks. By integrating RTT variations and MSS, the proposed method improves throughput, reduces latency, and optimizes network bandwidth utilization under various packet error conditions. These results highlight MRVHS-CCA's potential for real-world adoption and further research into fairness and adaptability in diverse networking environments. But it still needs to be improved as follows:
(1) The MRVHS-CCA proposed in the article is mainly based on RTT changes and dynamic adjustment of MSS. The current experiment is limited to a single network scenario (single-link cellular to cloud environment). However, the adaptability of the algorithm has not been fully verified in different network topologies and multi-path transmission environments. Suggested improvements: Test MRVHS-CCA in different network environments (such as multi-path TCP, cellular switching, MEC) to verify its generalization ability. Study its applicability on different TCP stacks (such as QUIC, SCTP) to enhance its versatility.

(2) The article mainly compares the performance of MRVHS with other TCP variants (such as BBR, CUBIC, NewReno), but does not analyze the fairness of MRVHS in multi-user scenarios. Traditional TCP variants may cause unfair throughput allocation problems when sharing bandwidth. It is necessary to evaluate whether MRVHS can maintain stability in different TCP competition environments.

(3) In the experimental analysis, the colors are difficult to distinguish. Please improve the quality of the figures, such as Figures 9 and 11.

(4) In the conclusion, integrate the last two sentences "For future work, we aim to investigate the performance of the proposed protocols across diverse network topologies and varying traffic conditions. Furthermore, we aim to evaluate the fairness of MRVHS in comparison with state-of-the-art protocols." in Line 224-226.

Experimental design

In the experimental analysis, the colors are difficult to distinguish. Please improve the quality of the figures, such as Figures 9 and 11.

Validity of the findings

(1) The MRVHS-CCA proposed in the article is mainly based on RTT changes and dynamic adjustment of MSS. The current experiment is limited to a single network scenario (single-link cellular to cloud environment). However, the adaptability of the algorithm has not been fully verified in different network topologies and multi-path transmission environments. Suggested improvements: Test MRVHS-CCA in different network environments (such as multi-path TCP, cellular switching, MEC) to verify its generalization ability. Study its applicability on different TCP stacks (such as QUIC, SCTP) to enhance its versatility.

(2) The article mainly compares the performance of MRVHS with other TCP variants (such as BBR, CUBIC, NewReno), but does not analyze the fairness of MRVHS in multi-user scenarios. Traditional TCP variants may cause unfair throughput allocation problems when sharing bandwidth. It is necessary to evaluate whether MRVHS can maintain stability in different TCP competition environments.

---

## Round 0.2 · accepted · Accept

Reviewers agree you improved the manuscript as requested.

·

Basic reporting

It is well written.

Experimental design

The authors have conducted the experiment which is sufficient to check the validity of proposed algorithm.

Validity of the findings

Results are improved.

Additional comments

The authors have incorporated my comments and solved my query.

Reviewer 2 ·

Basic reporting

The author has addressed all of my queries; however, the Related Work section still lacks recent references from 2023 to 2025.

Experimental design

No further comments.

Validity of the findings

No further comments.

Additional comments

No further comments.

Reviewer 3 ·

Basic reporting

clearly modified

Experimental design

clear and author discussed

Validity of the findings

Author discussed and incorporated

Additional comments

Author addressed all the questions